# An Exploration of Student Nurses’ Experiences of Burnout during the COVID-19 Pandemic Using the Copenhagen Burnout Inventory (CBI)

**DOI:** 10.3390/healthcare11182576

**Published:** 2023-09-18

**Authors:** Charlie Cottam, Aimi Dillon, Jon Painter

**Affiliations:** Department of Nursing and Midwifery, Sheffield Hallam University, Sheffield S1 1WB, UK; a.dillon@shu.ac.uk (A.D.); j.painter@shu.ac.uk (J.P.)

**Keywords:** nurse, student nurse, burnout, COVID-19, employment, resilience

## Abstract

Burnout amongst healthcare professionals has been a long-considered condition associated with the workplace environment. Student nurses studying at Sheffield Hallam University continued to engage in their training during the COVID-19 pandemic; however, the stressors of this experience were anecdotally highlighted to their academic staff. Furthermore, burnout can be linked to the ongoing difficulties with recruitment and retention of nursing staff within the NHS workforce. This work aimed to determine the burnout among nursing students experience by obtaining quantitative data to understand their experiences. The Copenhagen Burnout Inventory was used to gauge levels of burnout across the different fields of nursing students. Results identified that: (1) mental health students reported feeling tired significantly less often than child and adult field students (mean rating of 69% versus 91.7% and 84.0%, respectively); (2) students aged 30–39 feel tired significantly less often than both younger student age groups (mean rating 59.4% versus 82.8% and 90.6%); (3) there was a significant difference in how often different age groups felt “*tired of working with clients*” (F(4) = 2.68, *p* = 0.04) and that “*they couldn’t take it anymore*” (F(4) = 2.53, *p* = 0.05); (4) child-field students reported generally higher levels of global burnout (mean CBI total = 57.9%) whilst mental health students reported lower levels (mean CBI total = 54.1%). Considering these results, it is imperative for both higher education institutions and potential employers to consider the impact of COVID-19 and burnout, and the levels of support offered to student nurses during their training and transition to practice as newly qualified nurses.

## 1. Background and Introduction

The concept of professional burnout was introduced in the 1970s by an America psychologist called Herbert Freudenberger [1]. Burnout is a condition linked with employment, characterised by a plethora of symptoms such as fatigue/exhaustion, reported negativity and a sense of ‘distance’ from the individual’s job, and a sense of role fatigue [2]. As the archetypal caring profession, nursing by definition involves a significant emotional burden, with burnout being an inherent occupational hazard [3].

Previous studies have shown differing levels of burnout between caring professions [4,5,6,7]. The reasons for these differences are not fully understood and are likely to be due to a complex and multi-faceted combination of each staff group’s unique challenges and characteristics. The Nursing and Midwifery Council (NMC) is the United Kingdom’s (U.K.) nursing regulatory body. According to the NMC‘s [8] annual report, the parts of its register with the largest memberships are adult nursing (587,885), mental health nursing (95,485), and child nursing (57,014). Given the heterogeneity of these fields of nursing practice, it is logical to expect similar variations in the levels and nature of burnout.

In the U.K., the incidence of nursing burnout in general has been exacerbated by the additional demands that COVID-19 created [9,10]. Already stretched services experienced increased demand upon resources as well as increased staff vacancies across the National Health Service (NHS). As of mid-2022, the vacancy rate for the NHS workforce (Registered Nursing category) was 38,972—an increase of 9.2% on the previous year of 34.678 [11]. Vacancy figures for the current financial year are yet to be released but the NHS long term workforce plan [12] is predicated on the shortfall continuing for years to come. The plan therefore outlines a “near doubling of nurse training places” (p. 49) and a range of initiatives to improve retention rates of nurses once trained (p. 58). These specifically include focusing on retention during the early stages of nursing careers (p. 79) including preceptorship (p. 80).

In many ways, therefore, student nurses can be viewed as crucial to the success of the NHS’s workforce plan. Sheffield Hallam University (SHU) is the sixth largest leading provider of further education in the U.K. [13] and one of the largest providers of pre-registration nurse education. In the U.K., nurse training is 50% classroom learning and 50% supervised clinical practice. This equates to 2300 h spent working in these highly pressured clinical environments, which is higher than many other comparable countries. Through COVID-19, there were anecdotal reports that rates of calls to SHU’s student wellbeing helpline increased dramatically, with nursing students experiencing high levels of stress reportedly brought on by the transition of their teaching to online modes, as well as having to learn the practical elements of the role in clinical services exposed to and struggling to cope with the pandemic. There is some international evidence of COVID-induced student nurse burnout, e.g., from Iceland [14], China [15], and Japan [16]. However, Mulyadi et al.’s [17] systematic review identified a dearth of empirical evidence from the UK. This led to the current study into the experiences of students who trained during the pandemic and are about to enter the UK’s already stressed NHS workforce.

### 1.1. Aim

The aim of this study was to determine the levels of burnout that were experienced by third-year nursing students due to enter the NHS workforce in September 2023.

### 1.2. Research Questions

(1)What levels of burnout are reported by third-year nursing students due to graduate in September 2023?(2)Do the levels of burnout affect all nursing students equally?

## 2. Method

### 2.1. Study Design

Our study was cross-sectional in nature with three field-specific groups of third-year students in the same cohort asked to complete an online questionnaire at the same point in their timetabled sessions.

### 2.2. Participants

From the 552 final (third) year nursing students due to complete their course in September 2023, a convenience sample of 55 students agreed to participate in the study. Of these, 51 were female and the median age category was the 21–29-year-old bracket. Table 1 shows the distribution of participants by field of practice as well as their demographics.

### 2.3. Measure

This study utilised the Copenhagen Burnout Inventory (CBI) [18] which is a 19-item measure, with each item having a 5-point rating ranging from “Never/Almost Never” or “To a Very Low Degree” (0%); “Seldom/To a low degree (25%); Sometimes/Somewhat (50%); Often/To a high degree (75%) through to “Always”/“To a Very High Degree” (100%) (see Table 2). The items are grouped under three theoretically derived subheadings: personal burnout (n = 6), work-related burnout (n = 7), and client-related burnout (n = 6). Totals for each subscale are reported as the average percentage rather than a cumulative total. It was produced in Denmark, in part due to dissatisfaction with the Maslach Burnout Inventory [19]. The tool was developed using data from participants (n = 1914) working in various service industries including psychiatric hospitals and prisons, social welfare offices, general hospital wards, learning disability institutions, and homecare services. It was subsequently shown by Campos et al. [20] to be valid for use in a student population. During development and validation, the CBI proved easy to understand, had high response rates, and very high internal consistency (with Cronbach’s alphas between 0.85 and 0.87). Convergent validity between the CBI and other validated measures was also demonstrated. Correlations between the three latent constructs were 0.72, 0.46, and 0.61. The tool also showed sensitivity to change over time and clinically intuitive associations with aspects of job satisfaction. In summary, therefore, although the Maslach Burnout Inventory has been used more widely, the CBI is a robust and well-validated tool [21].

### 2.4. Recruitment and Procedures Undertaken

The study was advertised to all third-year adult, child, and mental health student nurses using the university’s global email and module announcement systems. Promotional flyers were also placed throughout the campus. Students that expressed an interest were supplied with participant information sheets and given the opportunity to ask the researchers questions about the study prior to recording their consent on the mandatory first screen of an online Qualtrics [22] questionnaire. Participants were then invited to record basic demographic information (age banding, gender, and field of nursing) prior to completing the CBI itself. Following completion of the tool, participants were provided with details of appropriate student support services in the event that answering the survey had caused any distress. These data were gathered over three separate (field-specific) sessions during May 2023.

### 2.5. Data Analysis

The anonymised survey responses were exported from Qualtrics into SPSS version 26 [23] for analysis. All data were stored securely on the university’s encrypted research servers. As per the original rating guidance, responses were translated into percentages with “Never/Almost Never” and “To a Very Low Degree” = 0%; “Seldom” and “To a low degree” = 25%; “Somewhat and Sometimes” = 50%; “Often” or “to a high degree” = 75%; and “Always” and “To a Very High Degree” = 100%. Note that Question WR4 (see Table 2) required the rating to be inverted due to the wording.

A check of the tool’s internal consistency was performed using McDonald’s ω (a more robust alternative to the traditional Cronbach’s alpha) before mean ratings (and standard deviations) for each question; the sub-scale and tool total were calculated for the entire sample, for each field of practice, and for each age range. Following this, data were checked for normality of distribution, homogeneity of variance and independence. Analyses of variance (ANOVA) were then conducted with post hoc Dwass–Steel Critchlow–Fligner pairwise analyses to understand more specifically where the statistically significant variances lay.

### 2.6. Ethical Approval

All participants were advised that participation was voluntary, and that no participant identifiable data would be reported. Informed consent was gathered via the opening page of the online survey. All aspects of this study complied with the ethical standards of the relevant national and institutional committees on human experimentation and with the Helsinki Declaration of 1975 (as revised in 2008). The study was ethically approved by Sheffield Hallam University (Review ID: ER52334360).

## 3. Findings

Our sample comprised 10.0% of the total cohort; however, the proportion varied by field as the adult field cohort was twice the size of the mental health and child-field cohorts combined. Therefore, we sampled 6.8% of adult field students, 10.7% of child-field, and 20.4% of mental health students.

Internal consistency of the CBI had a very good McDonald’s ω = 0.916, confirming all questions were aligned to the same underlying construct (i.e., burnout). Mean ratings for the 19 CBI questions ranged from 25.5% (CR5) to 79.5% (PB1) with the mean total CBI score being 56.2% (Table 3). Of the three subtotal mean ratings, personal burnout was highest (67.4%) closely followed by work-related burnout (64.1%) with the client-related subtotal being notably lower (35.7%). The child-field students’ ratings were highest for almost half of the questions, though it must be acknowledged that the sample was small in comparison to the adult and mental health samples. In terms of age, there was a fairly consistent pattern across all questions, sub-totals, and overall total, whereby mean ratings reduced from the 18–20-year-old range through to the 31–39-year-olds before rising again in the 40–49-year-old age bracket. The 50–59-year-old range tended to be relatively high; however, few conclusions can be drawn from this as it comprised a single student.

When considering the effect of the students’ field of practice, two questions were noteworthy. ANOVA (Table 4) found that responses to question PB1 varied significantly by field of practice F(2) = 5.12, *p* = 0.01. Post hoc pairwise comparisons (Table 5) found that the mean ratings were significantly lower for mental-health-field than child-field students: w = −3.35, *p* = 0.047. Question CR4 also varied significantly by field: F(2) = 3.56, *p* = 0.04; however, post hoc analysis did not reveal any noteworthy pairwise comparisons.

When considering the effect of the students’ age, three questions were noteworthy. ANOVA found that responses to question PB1 varied significantly by student age: F(4) = 3.22, *p* = 0.02. Post hoc pairwise comparisons found that the mean ratings were significantly lower for 30–39-year-olds than both 18–20 year olds (w = −4.11, *p* = 0.003) and 21–29 year olds (w = −3.9, *p* = 0.046). Question PB4 (F(4) = 2.53, *p* = 0.05) and CR5 (F(4) = 2.68, *p* = 0.04) also varied significantly by age; however, post hoc analysis did not reveal any noteworthy pairwise comparisons.

## 4. Discussion

Our findings show that students who have trained during the COVID pandemic have worryingly high levels of burnout. Frequency of tiredness varied significantly by both age and field of practice, with 30–39-year-olds and mental health students reporting the lowest frequencies. In addition, students from each field reported differing levels of “giving more than they got back from patients” whilst feeling “tired of patients” and as though they “couldn’t take any more” both varied significantly by age group.

Comparing our study to similar studies conducted internationally reveals that our students report higher levels of burnout. Montgomery et al.’s [4] study, for example, indicated very low levels of burnout generally, aside from point CR6 (CR6—*Do you sometimes wonder how long you will be able to continue working with clients?*) whereas our students generally felt that this was not an issue. Campos et al.’s [20] study explored the presence of burnout in Brazilian healthcare students, highlighting significant levels of burnout being reported across all domains of the CBI. Of course, these studies were undertaken pre-COVID.

Tiredness and burnout have long since been found to be strongly linked. For example, a study by Skorobogatova et al. [24] of Polish nurses revealed that tiredness and sleeplessness, among some other health complaints, were the most common complaint. Tiredness was identified as having the strongest links to burnout. Similarly, Membrive-Jiménez et al. [25] showed within their systematic review of burnout and sleep problems that a high proportion of nurses experiencing burnout also experienced poor levels of sleep.

Furthermore, it is globally recognised that the transition from student nurse to newly qualified nurse (NQN) is a notably difficult period [26]. During the transition to registered nurse, NQNs are at particular risk of suffering from the effects of stress and burnout [27,28]. Seminal work by Kramer [29] focused on NQNs’ experiences in the US. The term “reality shock” was coined, which Kramer used to describe a reaction that NQNs could have once they begin their new role. This can lead to stress and feelings of being overwhelmed, making the NQN more likely to leave their first role and/or leave the profession completely [29]. Further to this, Missen et al. [30] conducted a systematic review exploring job satisfaction in the NQN workforce. One of their findings concluded that the phenomenon of reality shock is still relevant.

Our research identified that younger students (18–29) felt tired statistically more often than students aged 30–39. Overall, our sample reported the frequency of tiredness at higher levels than Montgomery et al.’s [4] study of US registered nurses (61.8% versus 43.66%). However, the ranked position of this question was very similar in both studies (seventh versus eighth out of the 19 questions). In addition to the additional burden of in vivo learning (versus practicing as a nurse with more experience and competence), as discussed above, this could also be because practicing during the pandemic was generally more stressful.

Our 30–39-year-olds reported the lowest levels of tiredness; however, the average age of our participants was lower than Skorobogatova et al.’s [24], which was 41.8. However, both studies found strong links between tiredness and burnout. The reasons for this are unclear; however, elsewhere, this age range, described as generation Y (born 1980–1994), have been found to feel that if work-related fatigue becomes a problem, then alternative employment will be sought [31]. It may be, therefore, that by their final year, members of generation Y who have experienced work-related tiredness have already left the course. Either way, this potential loss of workforce due to fatigue must be borne in mind by higher education institutions (HEIs) and future employers alike. The recently published NHS Long Term Workforce Plan [12] also highlights health and wellbeing, among other factors, as to the reasons why staff leave the NHS.

Stephens [32] developed the Stephens Model of Nursing Student Resilience in an attempt to tackle this issue of retention because of reality shock. Although resilience is recognised as a complex phenomenon, many authors believe that it is something that can be developed or enhanced [32,33,34]. The concept of resilience is increasingly discussed in relation to healthcare and healthcare professionals. HEE [27] notes workforce resilience as one of the biggest challenges faced by the NHS.

In addition to age, we also noted variation in the frequency of tiredness by field of practice. This echoes Hansen & Virden’s [35] study of healthcare students from different professions, where similar variability was evident in sub-section totals for the CBI. Our results indicate (within Table 3) that our child-field students are reporting higher levels of burnout than other branches. Various studies have been undertaken relating specifically to burnout within child nursing [36,37,38,39] and also considering potential factors which may contribute to a level of burnout, as well as potential methods to improve these. Maytum et al.’s [38] qualitative pilot project exploring the theme of compassion fatigue and burnout of nurses working with chronic illnesses in children and their families identified that those symptoms of compassion fatigue were similar to those experienced with burnout (although the severity of the reported symptoms was greater with burnout). They also identified that the presence of compassion fatigue potentially advances to a state of burnout, which is inevitably longer lasting.

By the nature of mental health nursing, it may be argued that mental health student nurses have a lower level of burnout present due to the fact that they may be more ‘psychologically equipped’ for the roles in which they are to be working. Mental health nursing students may also be more prepared for dealing with trauma; frequent exposure to highly distressing emotional experiences; and incidents of self-harm/suicide. Quigley et al.’s [37] study of associated burnout and quality improvement within paediatric nursing also highlights lower levels of burnout and innovations to improve patient and family care experiences.

Considering the context of resilience further, it is crucial that student nurses embrace the concept of self-compassion. Studies completed by Hashem & Zeinoun [40]) suggest that the development of self-compassion within a healthcare workforce can significantly interrupt the burnout developmental process. The role of the nurse is one that is always expanding with demand; Huhtala et al. [41] consider the impact of intensified job demands on wellbeing (and job satisfaction). In their study of over a thousand employees, they argue that work intensification and increased employee demands to plan and execute one’s own workload were two job demands that significantly impacted areas such as exhaustion, noting specifically nurses working within the field of emergency care provision. Mabala et al. [42] note, through qualitative research conducted in South Africa with newly qualified nurses, that there are a number of themes which impact nurse retention, including that without appropriate support systems in place, there is a risk of nurses failing to thrive. *The Future of NHS Human Resources and Organisational Development Report* [43] highlights a number of actions intended to support both staff and services, such as simplifying the way colleagues interact with each other and embedding a culture of staff wellbeing.

Variation by field was also evident in feeling that *patients took more than they gave back* (question CR4). Here, adult student nurses had the highest score (57%). Although we advocate person-centred care to all fields throughout their training, it is possible that adult nursing during the pandemic became more task orientated, meaning that the less obvious pastoral care was perceived to be secondary and unduly burdensome [44]. This might contrast with the mental health students for whom meeting patients’ emotional needs is always the mainstay of their role.

In this regard, mental health students may also be more attuned to their own emotional needs. Emotional intelligence (EI) is fundamental in helping to recognise one’s emotions and how to control them [45]. Developing EI is difficult, and self-awareness is just the first step to becoming more emotionally intelligent. It refers to understanding emotions before being engulfed by them [45] and this is a key part of burnout [33]. Furthermore Jacobson [3] discusses the need to control mindset in order to avoid burnout. Educating students to deliver the five steps to mental wellbeing [46] to their patient population may in turn have assisted with their own personal resilience.

### 4.1. Study Limitations

As with most research, there are a number of caveats to our findings. Convenience sampling, and the number of survey responses obtained, inevitably limit generalizability; however, the participants were broadly representative of a typical cohort of students at SHU. Similarly, the cross-sectional, quantitative nature of this study, together with the modest number of variables collected, precludes a deeper understanding of the reasons for individual responses being gained. That said, our overall findings chime with other published studies and provide a useful basis for further research into this phenomenon in our student body.

The phenomenon of burnout is not new and has been problematic for healthcare staff for many years. How, therefore, can we further adequately equip and prepare our future nurses with resilience and reduce the risk of burnout occurring? *Student Minds* [47] highlighted in their study of over 1000 university students that 30% of students felt a deterioration in their mental health since commencing their studies, with 59% of students highlighting that managing their finances was causing them stress ‘often’ or ‘all of the time’.

### 4.2. Recommendations for Practice and Future Research

Our initial findings suggest that there are several factors that need to be considered in relation to practice. SHU is currently undertaking a curriculum redesign: resilience within practice is not formally considered until the final year of their training. The results from our study suggest that this needs to be earlier within the curriculum in order to develop emotional intelligence and resilience at an earlier stage, thereby reducing the risk of burnout occurring prior to graduating. This in turn may reduce the levels of burnout being reported by our NQNs in practice.

Secondly, a period of preceptorship is vital to support NQNs when they transition into their new roles to reduce the risk of burnout. The NMC [48] and others [49,50] strongly recommend a period of preceptorship with protected learning and access to a preceptor. Preceptorship provides structure for new registrants with the aim of integrating them into the workforce [12]. Furthermore, the NMC’s [48] *Principles for Preceptorship* highlights that preceptorship should support NQNs’ mental health and wellbeing as well as helping to build confidence. Quek and Shorey [51] found that having access to a preceptor increased NQN confidence and the feeling of being part of a team, as well as reducing their anxiety.

## 5. Conclusions

We found worryingly high levels of burnout in students that had trained during the COVID pandemic. Both HEIs and future healthcare employers need to appreciate the levels of burnout which are experienced by third-year student nurses. Our study highlights that this is a very current and real issue facing our students and the newly qualified workforce. This year of their study is typically associated with more complex academic study (for example, writing their dissertations), as well as preparation for the transition to practice as registered nurses. It is therefore essential to identify any ‘at risk’ groups of student nurses who are at this point of transition. It is clear that further consideration needs to be given to the development of key leadership skills, such as resilience, at an earlier phase of their training to prepare them for the reality shock of today’s contemporary nursing practice.

## Figures and Tables

**Table 1 healthcare-11-02576-t001:** Study Participants.

Field of Nursing	Sample	Male	Female	18–20 yrs	21–29 yrs	30–39 yrs	40–49 yrs	50–59 yrs
Adult nursing students:	n = 25	n = 1 (4.0%)	n = 24 (96.0%)	n = 3 (12.0%)	n = 15 (60.0%)	n = 2 (8.0%)	n = 5 (20.0%)	n = 0 (0.0%)
Mental health nursing students:	n = 21	n = 3 (14.3%)	n = 18 (85.7%)	n = 2 (9.5%)	n = 8 (38.1%)	n = 6 (28.6%)	n = 4 (19.0%)	n = 1 (4.8%)
Children’s nursing students:	n = 9	n = 0 (0.0%)	n = 9 (100.0%)	n = 3 (33.3%)	n = 6 (66.7%)	n = 0 (0.0%)	n = 0 (0.0%)	n = 0 (0.0%)
Total:	n = 55	n = 4 (7.3%)	n = 51 (92.7%)	n = 8 (14.5%)	n = 29 (52.7%)	n = 8 (14.5%)	n = 9 (16.4%)	n = 1 (1.8%)

**Table 2 healthcare-11-02576-t002:** CBI questions.

PB1	How often do you feel tired?
PB2	How often are you physically exhausted?
PB3	How often are you emotionally exhausted?
PB4	How often do you think: “I can’t take it anymore”?
PB5	How often do you feel worn out?
PB6	How often do you feel weak and susceptible to illness?
WR1	Do you feel worn out at the end of the working day?
WR2	Are you exhausted in the morning at the thought of another day at work?
WR3	Do you feel that every working hour is tiring for you?
WR4	Do you have enough energy for family and friends during leisure time? (inverse scoring)
WR5	Is your work emotionally exhausting?
WR6	Does your work frustrate you?
WR7	Do you feel burnt out because of your work?
CR1	Do you find it hard to work with clients?
CR2	Does it drain your energy to work with clients?
CR3	Do you find it frustrating to work with clients?
CR4	Do you feel that you give more than you get back when you work with clients?
CR5	Are you tired of working with clients?
CR6	Do you sometimes wonder how long you will be able to continue working with clients?

**Table 3 healthcare-11-02576-t003:** Mean ratings and standard deviations for the CBI.

	Field/Age	PB1	PB2	PB3	PB4	PB5	PB6	PB Mean	WR1	WR2	WR3	WR4	WR5	WR6	WR7	WR Mean	CR1	CR2	CR3	CR4	CR5	CR6	CR Mean	Total Mean
**Mean (SD)**	**Adult**	84 (17.5)	69 (18.1)	71 (21.3)	60.0 (27.0)	73.0 (16.0)	46.0 (20.0)	67.2 (14.0)	80.0 (22.8)	73.0 (17.6)	54.0 (25.7)	55.0 (27.0)	73.0 (19.0)	57.0 (19.8)	69.0 (19.5)	64.4 (10.3)	38.0 (24.1)	43.0 (31.9)	38.5 (26.6)	57.0 (23.4)	29.0 (21.3)	31.0 (30.0)	39.3 (20.5)	57.4 (11.2)
	**Child**	91.7 (12.5)	77.8 (19.5)	83.3 (28.0)	69.4 (16.9)	72.2 (19.5)	66.7 (28.0)	76.9 (14.7)	80.6 (16.7)	83.3 (17.7)	55.6 (16.7)	61.1 (28.3)	72.2 (29.2)	44.4 (20.8)	75 (17.7)	64.3 (14.9)	33.3 (21.7)	36.1 (28.3)	30.6 (20.8)	38.9 (28.3)	19.4 (16.7)	30.6 (27.3)	31.5 (18.9)	57.9 (11.1)
	**Mental Health**	69 (24.9)	63.1 (26.9)	70.2 (28.1)	60.7 (31.2)	67.9 (26.4)	51.2 (31.1)	63.7 (24.2)	75.0 (15.8)	63.1 (26.9)	58.3 (21.4)	51.2 (24.3)	72.6 (20.8)	58.3 (24.2)	69.0 (22.2)	63.6 (13.8)	33.3 (24.2)	36.9 (20.3)	33.3 (18.3)	40.5 (21.6)	23.8 (23.0)	32.1 (27.5)	33.3 (18.5)	54.1 (14.3)
	**Total**	79.5 (21.6)	68.2 (22.63	72.7 (25.1)	61.8 (27.2)	70.9 (20.8)	51.4 (26.5)	67.4 (18.9)	78.2 (19.3)	70.9 (22.4)	55.9 (22.5)	54.5 (25.9)	72.7 (21.1)	55.5 (21.9)	70 (20.1)	64.1 (12.3)	35.5 (23.4)	39.5 (27.1)	35.2 (22.5)	47.7 (24.7)	25.5 (21.2)	31.4 (28.1)	35.7 (19.4)	56.2 (12.3)
	**18–20 yrs**	90.6 (12.9)	78.1 (24.8)	84.4 (18.6)	62.5 (29.9)	78.1 (20.9)	62.5 (29.9)	76.0 (17.5)	75.0 (18.9)	84.4 (18.6)	50.0 (18.9)	62.5 (13.4)	84.4 (18.6)	43.8 (17.7)	65.6 (18.6)	62.9 (10.3)	37.5 (26.7)	37.5 (32.7)	34.4 (29.7)	56.3 (22.2)	21.9 (20.9)	37.5 (35.4)	37.5 (23.1)	59.0 (13.3)
	**21–29 yrs**	82.8 (20.2)	67.2 (19.0)	72.4 (24.4)	58.6 (24.3)	71.6 (16.0)	50.0 (22.2)	67.1 (15.0)	81.9 (19.9)	73.3 (17.6)	56.0 (21.8)	56.0 (29.6)	69.0 (21.8)	58.6 (21.4)	75.0 (16.4)	65.4 (11.7)	37.9 (21.8)	45.7 (27.6)	39.7 (20.6)	46.6 (25.6)	29.3 (19.0)	32.8 (29.2)	38.6 (18.6)	57.5 (11.0)
	**30–39 yrs**	59.4 (18.6)	56.3 (29.1)	53.1 (28.1)	46.9 (31.2)	59.4 (29.7)	37.5 (29.9)	52.1 (23.5)	71.9 (16.0)	56.3 (32.0)	46.9 (20.9)	43.8 (29.1)	75.0 (26.7)	56.3 (29.1)	59.4 (29.7)	60.3 (17.9)	25.0 (29.9)	25.0 (26.7)	25.0 (26.7)	40.6 (22.9)	21.9 (28.1)	28.1 (28.1)	27.6 (24.7)	47.4 (15.2)
	**40–49 yrs**	75.0 (18.6)	69.4 (20.8)	80.6 (24.3)	80.6 (20.8)	72.2 (26.4)	55.6 (32.5)	72.2 (21.7)	75.0 (21.7)	63.9 (25.3)	66.7 (28.0)	52.8 (19.5)	72.2 (15.0)	52.8 (19.5)	66.7 (21.7)	63.5 (11.9)	33.3 (21.7)	33.3 (17.7)	28.1 (16.0)	47.2 (26.4)	13.9 (13.2)	22.2 (19.5)	29.4 (11.4)	55.7 (11.1)
	**50–59 yrs**	100 *	100 *	75.0 *	100 *	75.0 *	75.0 *	87.5 (13.7)	75.0 *	75.0 *	75.0 *	50.0 *	75.0 *	75.0 *	75.0 *	71.4 (9.4)	50.0 *	50.0 *	50.0 *	75.0 *	75.0 *	50.0 *	58.3 (12.9)	72.4 (24.0)

* S.D. not applicable as only a single participant in age band.

**Table 4 healthcare-11-02576-t004:** Analyses of variance in burnout ratings.

CBI Question	Mean	S.D.	Variance by Field of Practice			Variance by Age		
			F	df	Sig.	F	df	Sig.
**PB1**	79.55	21.57	5.12	2	0.01	3.22	4	0.02
**PB2**	79.55	21.57	1.42	2	0.25	1.56	4	0.20
**PB3**	72.73	25.13	0.96	2	0.39	2.01	4	0.11
**PB4**	61.82	27.15	0.42	2	0.66	2.53	4	0.05
**PB5**	70.91	20.84	0.36	2	0.70	0.87	4	0.49
**PB6**	51.36	26.54	2.09	2	0.13	1.19	4	0.33
**PB Mean**	67.42	18.86	1.57	2	0.22	2.40	4	0.06
**WR1**	78.18	19.28	0.46	2	0.64	0.59	4	0.67
**WR2**	70.91	22.45	2.96	2	0.06	2.02	4	0.11
**WR3**	55.91	22.55	0.21	2	0.81	1.16	4	0.34
**WR4**	54.55	25.95	0.46	2	0.64	0.56	4	0.69
**WR5**	72.73	21.12	0.00	2	1.00	0.86	4	0.50
**WR6**	55.45	21.89	1.40	2	0.25	0.96	4	0.44
**WR7**	70.00	20.07	0.33	2	0.72	1.20	4	0.32
**WR Mean**	64.09	12.28	0.03	2	0.97	0.37	4	0.83
**CR1**	35.45	23.42	0.26	2	0.77	0.59	4	0.67
**CR2**	39.55	27.08	0.37	2	0.69	1.13	4	0.35
**CR3**	35.19	22.53	0.52	2	0.60	1.00	4	0.42
**CR4**	47.73	24.66	3.56	2	0.04	0.71	4	0.59
**CR5**	25.45	21.24	0.76	2	0.47	2.68	4	0.04
**CR6**	31.36	28.14	0.01	2	0.99	0.47	4	0.76
**CR Mean**	35.73	19.44	0.01	2	0.99	1.11	4	0.36
**Total CBI Mean**	56.21	12.34	0.51	2	0.61	1.73	4	0.16

**Table 5 healthcare-11-02576-t005:** Post hoc analyses of burnout ratings using Dwass–Steel Critchlow–Fligner pairwise comparisons (with statistically significant results asterisked).

	Pairs		W	p
**Pairwise comparisons for question PB1 by field of practice**	Adult	Child	1.58	0.504
	Adult	Mental Health	−2.98	0.089
	Child	Mental Health	−3.35	0.047 *
**Pairwise comparisons for question PB1 by age group**	18–20 yrs	21–29 yrs	−1.3	0.889
	18–20 yrs	30–39 yrs	−4.11	0.03 *
	18–20 yrs	40–49 yrs	−1.87	0.678
	18–20 yrs	50–59 yrs	1	0.955
	21–29 yrs	30–39 yrs	−3.9	0.046 *
	21–29 yrs	40–49 yrs	−1.21	0.914
	21–29 yrs	50–59 yrs	1.35	0.876
	30–39 yrs	40–49 yrs	1.73	0.739
	30–39 yrs	50–59 yrs	2.33	0.467
	40–49 yrs	50–59 yrs	1.36	0.872

## Data Availability

The exact data can be obtained from the corresponding author.

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
