# Peer review of "An Exploration of Student Nurses’ Experiences of Burnout during the COVID-19 Pandemic Using the Copenhagen Burnout Inventory (CBI)"

_healthcare, 2023, doi:10.3390/healthcare11182576_

Round 1
Reviewer 1 Report
First of all, thank you for giving me the opportunity to review Charlie Cottam's manuscript, entitled "An Exploration of Student Nurses' Experiences of Burnout during the COVID-19 pandemic using the Copenhagen Burnout Inventory Copenhagen Burnout Inventory (CBI)
Comments and Suggestions for Authors:
-Authors should provide numerical and statistical data in the abstract to support their findings.
-Authors should expand the introduction by commenting on the background of burnout as well as its current situation among nursing students. It would be interesting to include some reference to burnout in student nurses during the COVID-19 pandemic. Authors should comment in more depth on what the nursing curriculum in the UK is like and how it differs from that of other countries.
- Authors should specify whether they have performed any calculations to determine the sample size. If they have not, they should explain why.
-The authors should better explain whether the questionnaire used has been previously validated in the population studied. Authors should explain what the ranges of each subscale and total are, as well as their interpretation.
-Table 2 does not provide any data and is therefore recommended to be deleted.
-Authors should specify the type of sampling used.
-Authors should expand on the statistical analysis by providing more information on the tests performed: description of the sample, compliance with normality, comparison of variables,...
-Authors should explain why they used the McDonald's ω
-Tables 3, 5 and 4 are not very descriptive of the results. The authors should redo them, making them more readable. Table 4 should provide means and SD
-The discussion should start with a brief summary of the results obtained. The authors should provide a reasoned justification of the results obtained.
-Authors should check the bibliographical references, as they are not cited according to the journal's recommendations.
Moderate editing of English language required
Author Response
Dear Reviewer,
Thank you for providing us with your reviewers’ helpful comments. We have diligently addressed each one in turn. Mindful of your timescales, we have tabulated comments and responses below to aid your review.
We trust you feel the article is strengthened and now suitable for publication.
Kind Regards
Charlie Cottam, Aimi Dillon & Jon Painter.
Response to reviewer's comments:
|
Reviewer |
Comments |
Author's response |
|
1 |
Authors should provide numerical and statistical data in the abstract to support their findings |
Numerical and statistical data added as requested. |
|
Authors should expand the introduction by commenting on the background of burnout as well as its current situation among nursing students. It would be interesting to include some reference to burnout in student nurses during the COVID-19 pandemic. Authors should comment in more depth on what the nursing curriculum in the UK is like and how it differs from that of other countries. |
Background section developed as suggested to highlight the dearth of UK studies into student nurse burnout. |
|
|
Authors should specify whether they have performed any calculations to determine the sample size. If they have not, they should explain why. |
Thank you for raising this point and giving us the opportunity to clarify that, in line with the advice of Althubaiti (2023) regarding non-probability sampling, we did not undertake an a priori power calculation. Given the limitations of our study (which we acknowledge in the article) we have not sought to portray our results as generalisable to a wider population. |
|
|
The authors should better explain whether the questionnaire used has been previously validated in the population studied. Authors should explain what the ranges of each subscale and total are, as well as their interpretation. |
We have added to the detail of the tool’s original validation in healthcare settings to confirm that is has subsequently been shown to be valid for use in student populations. We have also explained the scoring for items and sub-scales. To our knoweldge, there are no published normative values for each subscale however, to aid reader’s interpretation of results, our discussion section makes comparisons between our findings and other published studies. |
|
|
Table 2 does not provide any data and is therefore recommended to be deleted. |
We appreciate this point however there is some discrepancy between reviews with reviewer 2 asking us to refer to this table instead of including wording of questions in the results text. We have therefore elected to preserve the table and make better use of it to ensure it is more meaningful. |
|
|
Authors should specify the type of sampling used. |
We have specified in the second line of section 2.1 that this was a convenience sample |
|
|
Authors should expand on the statistical analysis by providing more information on the tests performed: description of the sample, compliance with normality, comparison of variables,.. |
In response to this point, and a corresponding comment from reviewer 2, we have added detail of the analyses to this section and moved details about the nature of the sample to section 2.2 and table 1. |
|
|
Authors should explain why they used the McDonald's ω |
We have clarified that the McDonalds’s ω was used as data frequently violates assumptions for Cronbach’s alpha and, in these situations McDonalds’s ω is more reliable (McNeish, 2018 and Hayes, 2020) |
|
|
Tables 3, 5 and 4 are not very descriptive of the results. The authors should redo them, making them more readable. Table 4 should provide means and SD |
Table 3: revised to be more readable and take account of reviewer 2’s specific instruction. Table 4: means and SDs included together with other changes requested by reviewer 2. Table 5: statistically significant results made more obvious and pairwise comparisons with no significant results whatsoever deleted to aid readability |
|
|
The discussion should start with a brief summary of the results obtained. The authors should provide a reasoned justification of the results obtained. |
Brief summary of findings now included prior to discussing the potential reasons |
|
|
Authors should check the bibliographical references, as they are not cited according to the journal's recommendations |
Following your positive initial review, these have now been changed to the required ACS format |
For more details, please see the revised manuscript.
Reviewer 2 Report
Overview and general recommendation:
This is a study that reaffirms the need to delve deeper into the degree of stress suffered by nursing students during their training period. In this case, it focuses on the COVID-19 pandemic period and the division of the participants by nursing fields is interesting. Indeed, the study is quite influenced by limitations, especially the number of responses and socio-demographic variables, but it is positive to have found significant results.
In my opinion, despite the limitations of the study, there are several areas of improvement throughout the manuscript.
Major comments
-Both References and the bibliographic citations are not correctly referenced according to Healthcare style. A thorough revision is needed.
-There is no Design subsection in the Methodology. The design of this study should specified. Besides, the items of the STROBE checklist for cross-sectional studies must be met (https://www.strobe-statement.org/checklists/).
Minor comments
- All pages: Please remove the many double spaces in the text. Also, please check the page headings, as we are in the year 2023 and sometimes the page numbering does not work correctly.
- Lines 54-55: It is not necessary to add the university's teaching quality objectives, as this is the aim of all universities.
- Line 63: It is not necessary to use two verbs for the objective of the study. Besides, it is possible that the verb "to determine" is more appropriate for the study.
- Lines 71-77 and Table 1: This section should include the general profile of the participants (third-year nursing students), not their statistical results, which should go in the Results section.
- Line 74: Why is there a range in the median value? It should be just one value.
- Table 1: It is recommended to include here the descriptive statistics for the variables age and sex.
- Line 85: Missing a comma in the thousands.
- Line 103: Please replace "gender" with "sex", as this study refers to the biological variable (for more information, see "Sex and Gender in Research": https://www.mdpi.com/journal/healthcare/instructions#ethics).
- Lines 106-107: In the results, there is a missing piece of information: Did the online questionnaire not have any mandatory items?
- Lines 109-120: The statistical tests should be added. Therefore, references to them can deleted in the Results section.
- Lines 119-120: It is not necessary to mention Table 2 again.
- Lines 129-130: This information is not relevant to the study.
- Line 134: Please replace "(see table 3)" with "(Table 3)".
- Lines 145-146, 149-150, and 157: Table 2 contains the description of the items, so it is not necessary to include them in the text (the codes are sufficient).
- Table 3: The rows with "N" in the first column can be deleted, as this information can provided in Table 1. In addition, they are very redundant data and make Table 3 difficult to read.
- Table 4: Please replace the Chi-square statistic with the F-statistic.
- Lines 307-316: The conclusions should also specify the results obtained in the study itself.
- Line 318: All authors' contributions should be added using the CReDiT system. For example, who carried out the statistical analysis and who was responsible for data collection?
- Lines 319, 323-325: All these sections must be completed.
I hope my comments will help you to improve the manuscript.
Best regards.
Some minor spelling errors, especially in numerical values.
Author Response
Dear Reviewer,
Thank you for providing us with your reviewers’ helpful comments. We have diligently addressed each one in turn. Mindful of your timescales, we have tabulated comments and responses below to aid your review.
We trust you feel the article is strengthened and now suitable for publication.
Kind Regards
Charlie Cottam, Aimi Dillon & Jon Painter.
Response to reviewer's comments:
|
Reviewer |
Comments |
Author's response |
|
2 |
This is a study that reaffirms the need to delve deeper into the degree of stress suffered by nursing students during their training period. In this case, it focuses on the COVID-19 pandemic period and the division of the participants by nursing fields is interesting. Indeed, the study is quite influenced by limitations, especially the number of responses and socio-demographic variables, but it is positive to have found significant results. In my opinion, despite the limitations of the study, there are several areas of improvement throughout the manuscript. |
Thank you for confirming the merit of our work, despite its limitations. We trust that by taking account of your helpful feedback the article is now worthy of publication. |
|
Both References and the bibliographic citations are not correctly referenced according to Healthcare style. A thorough revision is needed. |
Following your positive initial review, these have now been changed to the required ACS format |
|
|
There is no Design subsection in the Methodology. The design of this study should specified. Besides, the items of the STROBE checklist for cross-sectional studies must be met (https://www.strobe-statement.org/checklists/). |
Apologies – design section now added. |
|
|
All pages: Please remove the many double spaces in the text. Also, please check the page headings, as we are in the year 2023 and sometimes the page numbering does not work correctly. |
Thank you for pointing this out. Both were attributable to the journal’s standard template however we have corrected both errors. |
|
|
Lines 54-55: It is not necessary to add the university's teaching quality objectives, as this is the aim of all universities. |
Removed as requested |
|
|
Line 63: It is not necessary to use two verbs for the objective of the study. Besides, it is possible that the verb "to determine" is more appropriate for the study. |
Revised as per comment (and also in the corresponding sentence of the abstract) |
|
|
Lines 71-77 and Table 1: This section should include the general profile of the participants (third-year nursing students), not their statistical results, which should go in the Results section. |
Table 1 revised and sample size as a percentage of each cohort transferred to the results section. |
|
|
Line 74: Why is there a range in the median value? It should be just one value. |
Apologies, we have now clarified that this is the median age range of participants as students were not asked for their exact age, instead they were asked to indicate an age bracket. |
|
|
Table 1: It is recommended to include here the descriptive statistics for the variables age and sex. |
Table 1 revised to include these statistics |
|
|
Line 85: Missing a comma in the thousands. |
corrected |
|
|
Line 103: Please replace "gender" with "sex", as this study refers to the biological variable (for more information, see "Sex and Gender in Research": https://www.mdpi.com/journal/healthcare/instructions#ethics). |
Thank you for signposting us to this guidance. We can confirm that participants were asked to record their self-identified gender, rather than biological sex. |
|
|
Lines 106-107: In the results, there is a missing piece of information: Did the online questionnaire not have any mandatory items? |
We have clarified that only the consent questions were mandatory. |
|
|
Lines 109-120: The statistical tests should be added. Therefore, references to them can deleted in the Results section. |
Detail removed from results section and added to method section as requested. |
|
|
Lines 119-120: It is not necessary to mention Table 2 again. |
Deleted as requested |
|
|
Lines 129-130: This information is not relevant to the study. |
Removed. |
|
|
Line 134: Please replace "(see table 3)" with "(Table 3)". |
Replaced as advised |
|
|
Lines 145-146, 149-150, and 157: Table 2 contains the description of the items, so it is not necessary to include them in the text (the codes are sufficient). |
All deleted as advised |
|
|
Table 3: The rows with "N" in the first column can be deleted, as this information can provided in Table 1. In addition, they are very redundant data and make Table 3 difficult to read. |
Deleted as requested |
|
|
Table 4: Please replace the Chi-square statistic with the F-statistic. |
Replaced as requested |
|
|
Lines 307-316: The conclusions should also specify the results obtained in the study itself. |
In response to reviewer 1’s request, a summary of our findings has been included at the start of the discussion sections. Therefore, to avoid repetition, we have made a more global statement in the conclusion. We hope this is acceptable. |
|
|
Line 318: All authors' contributions should be added using the CReDiT system. For example, who carried out the statistical analysis and who was responsible for data collection? |
Added |
|
|
Lines 319, 323-325: All these sections must be completed. |
Completed |
|
|
Some minor spelling errors, especially in numerical values. |
We have re checked spellings |
For more details, please see the revised manuscript.
Round 2
Reviewer 1 Report
NONE
NONE
Author Response
Dear Editor / Reviewer
Many thanks for your continued review of our submission. Please find attached a revised version with appropriate amendments and changes completed as suggested. I hope that this is satisfactory.
Many thanks and best wishes,
Charlie Cottam.
Reviewer 2 Report
Thank you for reviewing all the content. After a review of the changes, I can only recommend the following minor modifications related to the journal's guidelines:
- Lines 58, 66, 85, 100, 104, 104, 127, 147,159, 184, 217, 223, 227, 233, 276, 291, 314, and 324: It appears that some double spaces still exist.
- Lines 17-18: "vs" and "versus" appear. Please unify the criteria.
- Line 87: Please remove the bold in "Study".
- Lines 348-462: Please, in articles published in journals, remove the periods that exist just after the name of the journals. Also, the volume number should be in italics.
Best regards.
Author Response
Dear Editor / Reviewer
Many thanks for the continued review of our submission. Please find attached a revised version with amendments and suggestions completed. I hope that you find this to be of a satisfactory standard.
With very best wishes,
Charlie Cottam.